# Tissue-Specific Transcriptomes in the Secondary Cell Wall Provide an Understanding of Stem Growth Enhancement in *Solidago canadensis* during Invasion

**DOI:** 10.3390/biology12101347

**Published:** 2023-10-20

**Authors:** Yu Zhang, Zhongsai Tian, Jiaqi Shi, Ruoyu Yu, Shuxin Zhang, Sheng Qiang

**Affiliations:** Weed Research Laboratory, Nanjing Agricultural University, Nanjing 210095, China; zhangyu2013@njau.edu.cn (Y.Z.); 2018216004@njau.edu.cn (Z.T.); 2018116083@njau.edu.cn (J.S.); swimmy_snape@163.com (R.Y.); 1260880963@njau.edu.cn (S.Z.)

**Keywords:** invasiveness, transcriptome analysis, *Solidago canadensis*, secondary cell wall, vascular tissue development

## Abstract

**Simple Summary:**

Secondary cell wall (SCW) deposition during plant vascular system development has major impacts on resource delivery and mechanical support provision in plants, which play a key role in the successful invasion of alien plants. However, few studies have focused on the transcriptional regulators of secondary wall biosynthesis during the invasion of *Solidago canadensis*. In this study, we screened two typical native (US01) and invasive (CN25) *Solidago canadensis* populations with different stem morphologies and compared their transcriptomes in both the phloem and xylem of the stem. In total, 66,648 and 19,510 differential expression genes (DEGs) were identified in the phloem and xylem. Bioinformatics analysis indicated that secondary cell wall (SCW) biosynthetic processes were dramatically affected in vascular tissues; the invasive population is dedicated to cellulose production and reducing lignin. These characteristics are likely to improve the strength and extensibility of the SCW and ultimately improve *S. canadensis* stem growth. Our study presents a novel insight into this mechanism that explains the success of plant invasion: SCW-related gene transcription mediates the tissue-specific development of vascular tissue, which contributes to an enhancement in aboveground vegetative growth during the successful invasion of *S. canadensis*.

**Abstract:**

Invasive plants generally present a significant enhancement in aboveground vegetative growth, which is mainly caused by variation in secondary cell wall (SCW) deposition and vascular tissue development. However, the coordination of the transcriptional regulators of SCW biosynthesis is complex, and a comprehensive regulation map has not yet been clarified at a transcriptional level to explain the invasive mechanism of *S. canadensis*. Here, RNA sequencing was performed in the phloem and xylem of two typical native (US01) and invasive (CN25) *S. canadensis* populations with different stem morphologies. A total of 296.14 million high-quality clean reads were generated; 438,605 transcripts and 156,968 unigenes were assembled; and 66,648 and 19,510 differential expression genes (DEGs) were identified in the phloem and xylem, respectively. Bioinformatics analysis indicated that the SCW transcriptional network was dramatically altered during the successful invasion of *S.canadensis*. Based on a comprehensive analysis of SCW deposition gene expression profiles, we revealed that the invasive population is dedicated to synthesizing cellulose and reducing lignification, leading to an SCW with high cellulose content and low lignin content. A hypothesis thus has been proposed to explain the enhanced stem growth of *S. canadensis* through the modification of the SCW composition.

## 1. Introduction

Alien plant invasion is widely recognized as a serious threat to natural and managed ecosystems. Consequences include native species displacement [1,2], the modification of ecosystem primary functions [3], and yield losses in agricultural production [4]. In different habitats, invasive species have often been cited as superior competitors over native species because of their stronger vegetative growth [5,6], prolific reproduction ability [7], and tolerance to stressful environments [8]. Among these invasive characteristics, a plant’s ability to allocate aboveground biomass is the key trait that increases its capacity to capture and utilize more light resources [9,10]. The most invasive populations are observed to have a higher biomass allocation ability compared with the native populations [5,6,11].

Plant structure and function are in large part determined by the development of the vascular system [12]. Differentiation in vascular tissues from the vascular cambium follows two different developmental pathways to produce phloem and xylem [13]. The roles of phloem and xylem in plant growth have been extensively investigated. The xylem has a direct role in the delivery of water and mineral nutrients from roots to aerial tissues [14]; the phloem is involved in the transport of fixed carbon, as well as other nutrients, from photosynthetic to heterotrophic tissues [15,16]. In recent research, the phloem has also been shown to carry additional cargo, including phytohormones (auxin, gibberellins, cytokines, and abscisic acid) and signaling agents involved in plant biotic or abiotic stress responses [17,18,19]. This combination of resource supply and delivery of information allows the vascular system to engage in the coordination of developmental and physiological processes at the whole-plant level.

During vascular tissue development, secondary cell wall (SCW) deposition enables plants not only to build a strong system of (xylem) vessel-based transport for water and minerals but also to attain fiber-based mechanical support for the plant body [20]. The secondary walls are composed mainly of cellulose, hemicelluloses, and lignin. The biosynthetic pathways of these secondary wall components have been biochemically and genetically characterized in great detail. Extensive research has shown that three cellulose synthase (CesA) genes are required for cellulose synthesis in secondary walls in different species [21,22]. Most of the genes involved in lignin’s biosynthetic pathway have been isolated and functionally characterized [23]. Recent evidence suggests several genes participate in hemicellulose biosynthesis [24,25]. Although SCW biosynthetic genes have been biochemically and genetically characterized in great detail, little is known about the molecular mechanisms underlying the coordinated expression of these genes during stem growth. Recent preliminary studies have explained the complex processes in which NAC and MYB transcription factors turn on the secondary wall biosynthetic program [26,27,28,29,30]. However, the coordination of the transcriptional regulators of the secondary wall biosynthetic program is complex; no comprehensive map of regulation has been developed at a transcriptional level during plant invasion.

*Solidago canadensis* is native to North America and has become an exceptionally successful worldwide invader, as the main invasive species in mid-west Europe and most of Asia, Australia, and New Zealand. In recent years, there has been an increasing interest in the successful invasion of *S. canadensis*; these studies have focused on *S. canadensis*’ invasion mechanism to effectively prevent and control its use. It has often been cited as a superior competitor over native species given its prolific vegetative growth [31]. Further research suggests that, in addition to producing more biomass than native species, the principal reason for *S. canadensis*’s invasion success is that it can effectively allocate biomass to stems and leaves rather than to roots [32]. Recent research indicates that the tissue-specific development of the xylem enhances *S. canadensis*’s growth ability, which plays an important role in the competitiveness of this species during invasion [33]. However, our understanding of the actual causes underlying its invasive success is very limited.

In this study, to further clarify the regulation mechanism that controls SCW deposition during the vascular tissue development of *S. canadensis*, we present an RNA-seq analysis to explore genes that are strongly associated with stem biomass allocation evolution during its invasion. We first screened two typical native (US01) and invasive (CN25) populations with different growth abilities by phenotype. Then, the transcriptomes from the stem phloem and xylem were compared. Ultimately, the influence of gene expression profiles on SCW deposition and vascular tissue development was investigated. Together, these analyses revealed that the crosstalk between secondary-wall-related TFs and their downstream SCW biosynthesis genes leads to modifications of the SCW composition and vascular tissue development to enhance stem biomass allocation in *S. canadensis*.

## 2. Materials and Methods

### 2.1. Sampling of S. canadensis Populations

Each wild *S. canadensis* population was sampled from a field site that was chosen in the sector with the greatest plant abundance. At each site, the specific sampling plots were 10 m apart to minimize the probability of sampling clonal ramets of the same individual. At least five individuals were collected at each sampling plot. From each selected plant, seeds were collected and placed in paper envelopes, air-dried, and stored at 4 °C until further use.

### 2.2. Common Garden Experiments

Common garden experiments were established at Nanjing Agricultural University (Nanjing, Jiangsu, China (32°2′ N, 118°50′ E), March 2017). Seeds from 10 invasive populations (IN) and 10 native populations (NA) of *S. canadensis* (Table 1) were germinated in 5 cm diameter plastic cups (300 mL) containing potting mix. The seedlings were then transplanted into 13 cm diameter pots (600 mL) with 4 seedlings per pot. The seedlings were grown in a glasshouse at a temperature of 20–25 °C. After 60 days, the morphological characteristics, i.e., plant height (PH), stem diameter (SD), root length (RL), and leaf area (LA), of each population were first measured; the plants were then harvested and oven-dried (70 °C, 48 h). Thereafter, the dry weight of stem (DWS), dry weight of root (DWR), and dry weight of leaf (DWL) of each population were determined separately. In order to further explore the growth mechanism of *S. canadensis*, two representative populations, US01 and CN25, which exhibit significant growth differences, were sampled for RNA-seq analyses, and subsequent experiments were conducted to compare plant growth abilities.

### 2.3. Stem Biomass Allocation and Anatomical Analysis

To compare stem biomass allocation between the two populations, 3 cm long stem sections from the middle of plants (nine individual plants per population) were separated, oven-dried (70 °C), and ground with a hammer to pass through a 1 mm screen. The cell wall residue (CWR) content was determined after a two-stage dry matter extraction in ethanol and water [34]. The lignin content and cellulose content were determined as previously described [33]. Each experiment was repeated three times and performed with nine plants for each replicate.

For anatomical studies, a 1 cm long segment was sampled from the middle part of the plants (three individual plants per population) and was immediately placed in 70% ethanol/water (*v*/*v*). For each segment, 50 serial stem cross-sections, 100 μm thick, were prepared with a cryostat (Leica CM1950). Mäule staining was first performed. Stem cross-sections were incubated for 7 min in a 0.5% potassium permanganate solution and then rinsed 3–4 times with distilled water until the water solution was clear. Then, 15% HCl was quickly added until the deep brown color of the sections disappeared; all the 15% HCl solution was removed, and a concentrated ammonium hydroxide solution was immediately added. Wiesner staining was performed with phloroglucinol/HCl. After staining, sections were examined under a Carl Zeiss microscope AX10 (Carl Zeiss MicroImaging GmbH, Gottingen, Germany) and were digitalized as color images with a resolution of 10 μm per pixel.

### 2.4. RNA Extraction, cDNA Library Preparation, and RNA-Seq

For future RNA extraction, ten 3 cm long stem sections from the middle of the plants (ten individual plants per population) were sampled. The phloem and xylem of each stem were first separated with a knife and immediately stored in liquid nitrogen at −70 °C for further use. The total RNA of each sample with three biological replicates was isolated using a TRIzol™ Plus RNA Purification Kit (Invitrogen, Carlsbad, CA, USA) according to the manufacturer’s protocol. The RNA samples were sent to Genepioneer Biotechnology Co., Ltd. (Nanjing, China) for library preparation and sequencing with the HiSeq2000 platform from Illumina. To obtain clean data, raw pair-end reads with a 150 bp length were used for quality control by removing the adapter sequence and filtering low-quality reads (more than 20% of the bases with a quality score of less than 15).

### 2.5. De Novo Assembly and Functional Annotation

Given the absence of reference genomic sequences, de novo assembly was applied to construct transcripts with RNA-seq clean reads using Trinity. To construct a uniform transcriptome reference, all assembled transcripts from four libraries of *S. canadensis* were merged to generate reference transcripts and unigenes. The gene function was annotated based on the following databases: NR (NCBI non-redundant protein sequences); NT (NCBI non-redundant nucleotide sequences); Pfam (Protein family); KOG/COG (Clusters of Orthologous Groups of proteins); Swiss-Prot (a manually annotated and reviewed protein sequence database); KO (KEGG Ortholog database); GO (Gene Ontology).

### 2.6. Differentially Expressed Gene Analysis

For differentially expressed gene analysis, the Tophat software, version 2.1.1, was used to map the reads to the above-assembled transcript sequences. Each gene expression level obtained using the Cuffdiff software, version 2.1.0, was represented as fragments per kilobase of transcript sequence per million base pairs sequenced (FPKM). DEGs were identified with Cuffdiff and were required to have a 2-fold change (*p*-value ≤ 0.05).

### 2.7. Quantitative RT-PCR

The expression levels of 30 genes identified by DEG were validated via quantitative RT-qPCR performed on 7300 Real-Time PCR equipment (Applied Biosystems, Carlsbad, CA, USA). The cDNA was synthesized from 500 ng of total RNA using a PrimeScript 1st Strand cDNA Synthesis kit (TakaRa Bio, Dalian, China). *ScEF-1α* (F: 5′-AGACAAGCCACTCCGTTTAC-3′; R: 5′-CCATACCAGGCTTGATGATACC-3′) was used as an endogenous control, as it was the housekeeping gene. RT-qPCR primers were designed using the Primer5 software, version 5.0 (Appendix A). RT-qPCR was performed using a SYBR R Premix Ex TaqTM kit (TakaRa Bio, Dalian, China). The relative expression of the selected genes was normalized using the 2^−ΔΔCt^ method. Correlations between RNA-seq data and qRT-PCR were analyzed using 30 representative genes. The R2 value was obtained by the ratio of log2 (FPKM) values from RNA-seq to the log2 (relative expression level) of the qRT-PCR values.

### 2.8. Statistical Analysis

Statistical analysis was conducted using the SPSS software (IBM SPSS Statistics 20, Chicago, IL, USA). One-way ANOVA was used to analyze the effect of origin (invasive vs. native). Differences between means were tested with Duncan’s test, and significance was determined at *p* < 0.05 or *p* < 0.01. The correlation between different parameters was analyzed using the SPSS software (IBM SPSS Statistics 20, Chicago, IL, USA). Drawings were made in Origin version 8.0 (Origin Lab Corporation, Northampton, MA 01060, USA).

## 3. Results

### 3.1. Invasive Population of S. canadensis Displayed a Stronger Aboveground Growth Ability

We compared the morphology of IN and NA populations (Figure 1a). The results showed that the PH (105.9%, *p* < 0.01), SD (23.3%, *p* < 0.01), and LA (159.4%, *p* < 0.01) of the IN populations were significantly increased compared with their values in the NA populations. However, RL was not significantly different between the IN and NA populations.

For dry weight (Figure 1b), the DWS and DWL of the IN populations were 115.1% (*p* < 0.01) and 94.5% (*p* < 0.01), significantly higher than those of the NA populations (*p* < 0.01), respectively, while non-significant differences were observed in the DWR. The results implied that the difference in total biomass between the two populations was mainly caused by variation in aboveground parts during invasion.

In order to further explore such differences, two representative populations, US01 and CN25, which showed significant growth differences (Figure 1c,d), were selected for divergent stem morphology investigations in common garden experiments and were used for RNA-seq analyses. The plant height and stem diameter of CN25 increased by 109.5% (*p* < 0.01) and 92.5% over US01, respectively (Figure 1d).

### 3.2. RNA-Sequencing and De Novo Assembly

In order to reveal the mechanism of the stem morphology evolution of the IN population of *S. canadensis* at the transcriptional level, we performed RNA-seq on xylems and phloems from native (US01) and invasive (CN25) populations, with three biological replicates. Using the Illumina Hiseq2500 platform, a total of 296.14 million (M) clean reads were achieved, with Q30 values ranging from 88.6% to 93.1% (Appendix A). Given the absence of reference genomic sequences, de novo assembly was applied to construct transcripts from these RNA-seq reads in the Trinity software, version 2.2.0 [35].

The assembly results showed that there were 438,605 transcripts and 156,968 unigenes with N50 values of 842 and 826, respectively. The average lengths of transcripts and unigenes were 721.66 bp and 718.84 bp, respectively. The length distribution showed that 47% of the length of transcripts or unigenes ranged from 300 to 500 bp, while 3.7% were longer than 2000 bp (Appendix A and Appendix A).

### 3.3. Functional Annotation

Unigene annotation was performed by BLAST searching (e-value ≤ 10−5) against the NR (NCBI non-redundant protein sequences), Pfam (protein family), COG (Clusters of Orthologous Groups), KOG (euKaryotic Ortholog Groups), Swiss-Prot (a manually annotated and reviewed protein sequence database), KEGG (Kyoto Encyclopedia of Genes and Genomes), and GO (Gene Ontology) databases (Appendix A). In total, 95,346 unigenes, accounting for 60.74% of all unigenes, were annotated, with 34,059 in COG, 38,003 in GO, 20,726 in KEGG, 46,563 in KOG, 50,707 in Pfam, 57,418 in Swiss-Prot, and 91,541 in NR databases, respectively.

We analyzed homologous species by comparing the unigene sequences to the NR database, and the results showed that the three most abundant unigenes were distributed in *Helianthus annuus*, *Lactuca sativa,* and *Cynara cardunculus* (Appendix A). All of them belonged to the Compositae family, implying gene sequence conservation in Compositae and a close relationship with these three species.

In the COG classification, we found most unigenes were classified under “amino acid transport and metabolism” in a much higher amount than the other terms (Appendix A). In the GO database, “metabolic process” and “cellular process” were dominant within the “biological process” category; “cell part” and “cell” were dominant in the “cellular component” category; and “catalytic activity” and “binding” were dominant in the “molecular function” category (Appendix A).

### 3.4. DEG Identification between CN25 and US01 Populations Shows Tissue-Specific Expression

To further clarify the differences between native and invasive *S. canadensis*, we dissected the stems of US01 and CN25 and conducted RNA-seq. In total, 19,510 and 66,648 differential expression genes (DEGs) were identified in the xylem and phloem, respectively. This result shows that there were more differences in the phloem between US01 and CN25 compared with the xylem. In addition, 9690 DEGs were detected in both the xylem and phloem, and 9820 and 56,958 DEGs were specifically detected in the xylem and phloem (Appendix A), implying similar and different regulation processes in the xylem and phloem.

Of the total DEGs, we found that more genes were down-regulated compared with the number of up-regulated genes in both the xylem and phloem (Appendix A). To investigate the function of DEGs, we mapped all DEGs to terms in the GO and KEGG databases. In total, 276 biological processes (Appendix A) and 7 pathways (Appendix A) were enriched in the phloem; meanwhile, 366 biological processes (Appendix A) and 17 pathways (Appendix A) were enriched in the xylem.

Of the enriched GO terms, most biological processes were similar in the phloem and xylem, such as “plant-type cell wall organization”, “lignin biosynthetic process”, and “cell wall organization”. Common KEGG terms were also found in the phloem and xylem, such as “phenylpropanoid biosynthesis” and “flavonoid biosynthesis”, which are involved in lignin biosynthesis. The pathway involving the highest number of unique genes was “starch and sucrose metabolism”. The enrichment analysis indicated that the invasive population mainly exhibited differences in SCW biosynthesis processes compared with the native species.

### 3.5. MYB Transcription Factors Are Likely to Be Regulators of SCW Development in S. canadensis

SCW synthesis is a crucial process during vascular tissue development; several MYB TFs have been described as being important for such a process [26,28,29]. In this study, we identified 14 MYB or MYB-like genes that were differentially expressed between the two populations (Figure 2a). Among these MYB genes, those encoding putative *ScMYB46/83* and *ScMYB61* homologs showed relatively higher expression levels in the xylem than in the phloem. In contrast, genes that only encoded putative *ScMYB52* and *ScMYB58* homologs presented notable expression levels in the phloem. Compared with US01, genes that encoded putative *ScMYB46/83* and *ScMYB61* were up-regulated in both the phloem and xylem of CN25. In contrast, *ScMYB58* was significantly down-regulated in CN25. Our results strongly suggest a potential tissue-specific SCW development between the two populations.

### 3.6. Cellulose Biosynthesis Genes Are Highly Expressed in Invasive Populations of S. canadensis

Starch and sucrose metabolisms have been described as important for SCW deposition or composition [21,22]. By mapping to the KEGG reference pathways, a total of 672 genes were assigned to the “starch and sucrose metabolism” pathway (ko00500); different expression levels between US01 and CN25 were observed for 294 genes in the phloem and for 106 genes in the xylem. These included genes encoding key enzymes for matrix polysaccharides in cell walls (Appendix A).

The majority of these genes were up-regulated in both the phloem and xylem of CN25, such as genes encoding galacturonosyltransferase (GAUT), fructokinase (FRK), UDP-glucuronate 4-epimerase (GAE), Beta-xylosidase (xynB), and Pectinesterase (PME). Similarly, genes involved in cellulose metabolism, including UDP-glucose 6-dehydrogenase 5, sucrose synthase, cellulose synthase, and endoglucanase, were also up-regulated in both the phloem and xylem of CN25 compared with US01. DEGs that putatively encode eight cellulose synthase A (*CesA*) groups are shown in Figure 2b. Among the two populations, PCW CesAs (*CesA1*-, *CesA2*-, *CesA3*-, and *CesA6*-related) were expressed relatively highly in the phloem, with a lower abundance in the xylem. In contrast, SCW CesAs (*CesA4*-, *CesA7*-, *CesA8*-, and *CesA9*-related) transcript levels were relatively high in xylem. Compared with US01, the expression levels of PCW CesAs in CN25 were relatively high in the phloem, with relatively low expression levels in the xylem, while the transcript levels of most of the SCW *CesA* genes were relatively high for CN25 in both the phloem and xylem (Figure 2b). The (tissue-specific) expression patterns of PCW CesAs and SCW CesAs between US01 and CN25 were distinct, and the much higher expression levels of SCW CesAs in both the phloem and xylem of CN25 seem to suggest the activation of cellulose production in CN25.

Other classes of cellulose synthase identified in this study were cellulose synthase-like genes (CSLs), which include *CSLB*, *CSLD*, *CSLE*, *CSLG,* and *CSLH* (Appendix A). The transcripts of most of these CSL genes were less abundant than those of CesAs; notable transcript levels were only observed for *ScCSLD3*, *ScCSLG2*, and *ScCSLG3*. For CN25, most CSL gene transcript levels were relatively low in both the phloem and xylem compared with US01, except for *ScCSLD3* and *ScCSLD5* in the phloem and *ScCSLD1* and *ScCSLD4* in the xylem. In contrast to the CesAs, most CSL gene transcript levels were relatively low for CN25 in both the phloem and xylem, except for *ScCSLD3* and *ScCSLD5*, which were highly expressed in the phloem, and *ScCSLD1* and *ScCSLD4*, which were highly expressed in the xylem.

### 3.7. Lignin-Related Genes Are Weakly Expressed in Invasive Populations of S. canadensis

In this study, both KEGG and GO analyses revealed differentially expressed patterns of lignin-related genes between the two populations (Appendix A). Of these DEGs (Figure 2c), we identified 71 genes encoding three enzymes of the general phenylpropanoid pathway, which showed comparable transcript levels in the phloem and xylem. However, most lignin-specific genes (117 genes encoding seven enzymes) were highly expressed in the xylem in contrast to the phloem, except for the gene encoding an F5H homolog (Unigene197095), which was only detected in the phloem with no notable expression in the xylem.

In both the phloem and xylem, the majority of these general and lignin-specific pathway genes were down-regulated in CN25, except for *ScHCT* and *ScCOMT*, which were mostly highly expressed in the phloem for CN25. We also detected differentially transcribed genes involved in lignin assembly, such as genes from LAC families. Most of them were down-regulated in CN25 compared with US01, except for *ScLAC4*, which was up-regulated in CN25. Combined with previous studies, this suggests that there was a strong inhibition of stem cell wall lignification in CN25.

### 3.8. Confirmation of Anatomical and Biochemical Traits of US01 and CN25

To verify the effects of the above transcription level differences on stem morphology, the anatomical stem structures of the two populations were compared, with a focus on vascular tissue development. Both the phloem and xylem sclerenchyma cell walls in CN25 were significantly thinner than that in US01 (Figure 3a–h), which indicated a strong enhancement of SCW development in CN25. A cell wall composition analysis revealed that the cellulose contents in CN25 were 8.0% (*p* < 0.05) and 15.1% (*p* < 0.05) greater compared with US01 in the phloem and xylem, respectively (Figure 3m). This is consistent with the observed up-regulation of genes related to the cellulose metabolic pathway in CN25.

Wiesner lignin staining in stem cross-sections exhibited typically intense red staining of the SCW in both the phloem and xylem of US01 (Figure 3i). In contrast, CN25 had only low lignin levels in tracheary elements and xylary fibers (Figure 3k).

Stem cross-sections were then stained with Maüle reagent to compare SCW development. After the application of Maüle reagent, all CN25 vessel elements reacted positively (Figure 3l), corresponding to a typical SCW that presents only mature vessel (MV) elements, while most primary cell walls of US01 protoxylem vessels (PVs) reacted negatively and had noticeably thinner cells, with a small number of vessel elements reacting positively (MV) in the xylem fibers (XFs) (Figure 3j). Unexpectedly, CN25 had lower proportions of lignified tissues compared with US01.

The decreased total lignin content in CN25 was confirmed using the AcBr method. The lignin contents were 26.5% (*p* < 0.05) and 18.3% (*p* < 0.05) lower compared with US01 in the phloem and xylem, respectively (Figure 3n). Taken together, our results showed significant changes in secondary wall development and secondary wall components during invasion. CN25 presents, therefore, highly cellulosic but reduced SCW lignification with notably well-developed xylem vessels.

### 3.9. qRT-PCR Validation of RNA-Seq Data

In order to verify the reliability of the RNA-Seq data, qRT-PCR analyses were conducted on the same RNA pools that had been previously used for next-generation sequencing. Thirty genes were first randomly selected for qRT-PCR assays (Appendix A). The gene expression ratios between CN25 and US01 obtained via RNA-Seq were compared with those obtained via qRT-PCR. The results obtained using the two techniques were highly correlated for both the phloem and xylem (R = 0.760 and 0.876, respectively) (Figure 4a,b).

The transcript levels of the eighteen selected genes—including three related to the auxin signaling pathway (*ScTIR1*, *ScARF1*, and *ScARF19*), two related to the ET signaling pathway (*ScERS* and *ScERF*), three related to MYB transcription factors (*ScMYB46*, *ScMYB83-1* and *ScMYB61*), three related to cellulose biosynthesis (*ScCesA4-1*, *ScCesA7-1*, and *ScCesA8-1*), and seven related to lignin biosynthesis (*ScPAL1*, *Sc4CL1*, *ScC4H*, *ScCCoAOMT1*, *ScCOMT1*, *ScCAD1* and *ScCCR1*)—were measured in the phloems and xylems of US01 and CN25 via qRT-PCR. The results showed similar expression patterns between RNA-seq and qRT-PCR, thus validating the RNA-seq data (Figure 4c,d).

## 4. Discussion

*S. canadensis* is a notorious invasive plant native to North America that is spreading across East China and has caused serious damage to the local agriculture and ecological environment. *S. canadensis* has shown excellent adaptability in China and has often been cited as a superior competitor over native species given its enhanced vegetative growth [31,36]. Additionally, studies have shown that most invasive plants are more developed aboveground compared with native plants regarding increases in competitiveness for light resources [6,11]. In this study, we observed a significant variation in stem phenotype between IN and NA populations. This finding is consistent with that of Szymura (2015), who suggested that stem growth enhancement is likely the main reason for the increased competitiveness of *S. canadensis*. However, there are few studies that focus on the relevant physiological and molecular mechanisms.

### 4.1. Tissue-Specific Transcriptome Remodeling Underlies the Drastic Differences between Stem Growth in US01 and CN25 Populations

In this study, through a transcriptome analysis, we successfully identified 156,968 unigenes in *S. canadensis*. This number of unigenes was 59.1% greater than previously reported for this species (98,643 unigenes) [37]. Via the differential expression analysis of genes in the US01 and CN25 populations, 66,648 and 19,510 DEGs were identified in the phloem and xylem, respectively. The differences in transcriptomes observed in our study were in agreement with the significant phenotypic variation between the two populations.

A function analysis showed that, in both the xylem and phloem, the DEGs between the US01 and CN25 were mainly involved in SCW biosynthetic processes, such as “plant-type cell wall organization”, “starch and sucrose metabolism”, and “lignin biosynthetic process”. All of these results indicate that SCW biosynthetic processes are dramatically affected during vascular tissue development and may have potential roles in the enhancement of vegetative growth during *S. canadensis* invasion.

### 4.2. Transcriptomic Networks Underlying the Up-Regulation of Cellulose Biosynthesis in the IN Population

During stem growth, vascular tissue development is a crucial process, endowing the stem with a variety of functions, such as mechanical support or water transport [12], and in turn, SCW deposition is a key process during vascular tissue development [13]. More recent studies have suggested that SCW cellulose synthesis is tightly regulated by MYB TFs via activin cellulose synthase (*CesA*) and cellulose synthase-like (*Csl*) genes [38,39]. MYB61 has yet to be confirmed as having a role in SCW development, especially in regulating cellulose synthesis in many species. Huang (2015) demonstrated that MYB61 could bind to and enhance the transcription of three secondary wall CesAs (*CesA4*, *CesA7*, *CesA8*); this regulatory machinery could increase cellulose contents and lead to rice stem elongation. Our results support such a function in *S. canadensis* stems. In our results, all SCW CesAs (*CesA4*-, *CesA7*-, and *CesA8*-related) were expressed more in the xylem compared with the phloem. The expression pattern of the *ScMYB61* gene was well correlated with the high cellulose content in xylems compared with phloems, which suggests that, as in other species where it strongly activates SCW-related *CesA* genes (Huang 2015), *ScMYB61* may specifically regulate *ScCesA* expression in *S. canadensis*. Compared with US01, the much higher expression levels of MYB61 and SCW CesAs in both the phloem and xylem could explain the higher cellulose content in CN25, indicating that the invasive population is ultimately dedicated to developing highly cellulosic SCWs in both the phloem and xylem (Figure 3). During SCW development, cellulose biosynthesis enhances a variety of mechanical properties in the cell wall, such as strength and rigidity, and consequently plays an essential role in many aspects of plant growth [40,41]. Many studies have shown that cellulose-deficient mutants exhibit a severe dwarfism phenotype and stem growth defects [42,43,44]. In our results, through *ScMYB61* regulation, cellulose biosynthesis in CN25 SCWs was enhanced simultaneously with xylem vessel development, which might be an important cause of the enhanced vegetative growth of *S. canadensis* during invasion.

### 4.3. Transcriptomic Networks Underlying the Down-Regulation of Lignin Biosynthesis in the IN Population

Previous studies have revealed that SCW lignification is significantly altered in invasive populations of *S. canadensis* [33]. Some reports have provided evidence that SCW lignification plays an important role during plant growth and development [12]. Lignin formation has been studied extensively as part of monolignol biosynthesis (the phenylpropanoid pathway) and polymerization (oxidization by peroxidases and laccases). However, the changes in phenylpropanoid pathway gene transcriptomes during plant invasion remain largely unknown. The present study revealed the global gene expression profiles of the phenylpropanoid pathway in IN and NA populations. The genes identified via RNA-Seq in this study mainly represent the main branches of the phenylpropanoid pathway (Figure 5). We generated a heatmap of gene expression ratios for each phenylpropanoid pathway gene and for each pair-wise comparison. The generated heatmaps were inserted into the lignin biosynthetic pathway (Figure 5). More interestingly, in both the xylem and phloem, most phenylpropanoid pathway genes were highly down-regulated in CN25 compared with US01. The low lignin content in CN25 in both the xylem and phloem can mainly be explained by the down-regulation of these lignin-specific pathway genes.

It was not anticipated that the transcript level of genes encoding *ScHCT* and *ScCOMT* would be higher in the phloem and more abundant in US01. It is possible that both HCT and COMT enzymes, if produced, are used in secondary metabolites rather than in monolignol production [45]. The phloem indeed contains flavonoids, and the expression of flavonoid-synthesis-related genes has previously been reported [18,46]. It is generally accepted that flavonoids have various functions in plant responses to abiotic and biotic stress [45]. In this study, the “flavonoid biosynthesis” process was enriched in the two populations (Appendix A), indicating that these pathways are likely to change after invasion, providing a direction for future research.

Previous studies have reported that MYB TFs can regulate the genes involved in lignin biosynthesis and affect lignin content during SCW development. *MYB58/63* has been shown to regulate phenylpropanoid pathway expression and, consequently, SCW lignification [28]. In this study, *ScMYB58/63* was down-regulated in the phloem, corresponding to a lower lignin content in CN25. However, notable expression levels were not observed in the xylem, indicating that there are other MYB TFs regulating the phenylpropanoid pathway in *S. canadensis* xylem.

Cell walls are complex structures in which the growing xylem cells confer the combined properties of strength and extensibility. Many studies have shown that secondary wall lignification forms cross-links between cell wall polymers, which imparts a “waterproof” property, as well as mechanical strength, rigidity, and environmental protection. However, these polymers also reduce cell wall extensibility, which hinders the elongation of cells [47,48]. In particular, Huang (2013) suggested that during organ elongation, increases in lignification to reinforce cell walls cause an inhibition in plant growth. It is, therefore, likely that the enhancement of stem growth in CN25 may be partly explained by a decrease in SCW lignification and is likely regulated through the MYB TF down-regulation of phenylpropanoid pathway genes (Figure 5). Such a potential regulatory network of lignin biosynthesis genes and MYB TFs described here may help us to understand the mechanism of stem growth enhancement in *S. canadensis* during invasion from the perspective of SCW development.

## 5. Conclusions

In conclusion, in comparison with the NA population, the IN population of *S. canadensis* exhibits significant differences in its transcriptional network regarding secondary wall deposition regulation and vascular tissue development. We proposed a hypothesis to explain the invasive mechanism that enhances *S. canadensis* stem growth through the modification of the SCW composition (Figure 6). Regulated through MYB TFs, the IN population is dedicated to cellulose production and reducing lignin, leading to SCWs with high cellulose contents and low lignin contents. These characteristics likely simultaneously improve the strength and extensibility of the SCW and ultimately improve stem growth. However, vascular tissue development responses during invasion are complex. We are still far from determining the full invasion response mechanisms of *S. canadensis*. Hence, the selection of optimum candidate genes is particularly important. The cellulose synthase A genes (*ScCesA4*, *ScCesA7*, and *ScCesA8*) identified in our study, which were highly expressed in both the phloem and xylem of the IN population and regulate vascular tissue differentiation, could be candidate genes for further investigations and genetic engineering.

## Figures and Tables

**Figure 1 biology-12-01347-f001:**
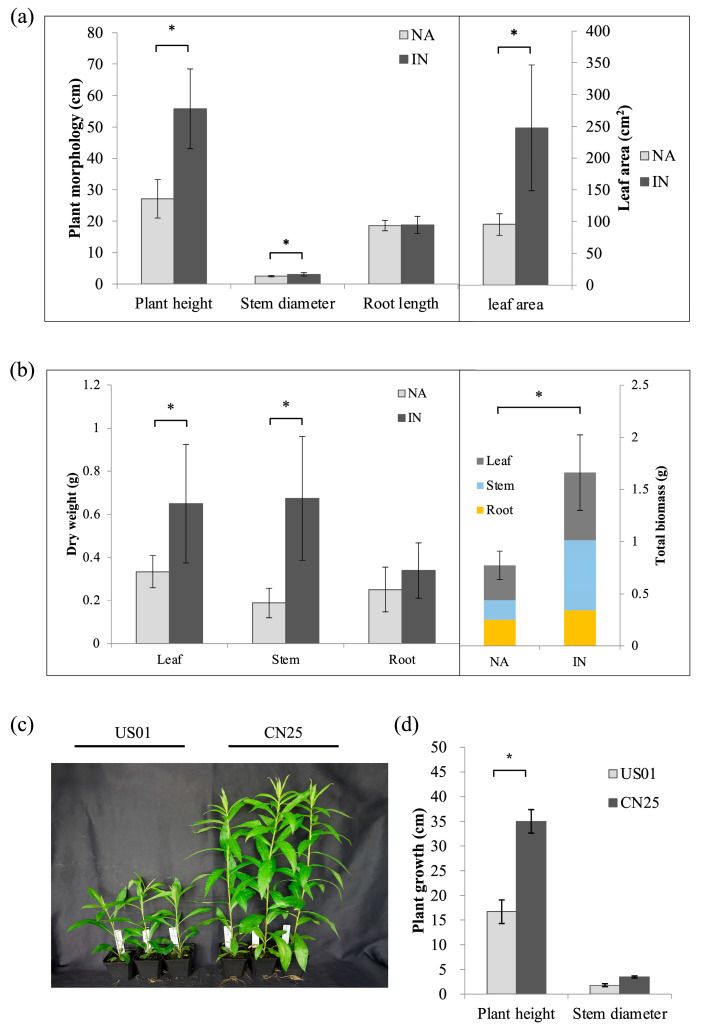
Comparison of morphological differences between invasive (IN) and native (NA) populations of *S. canadensis*. (**a**) Plant height, stem diameter, root length, and leaf area of 10 invasive and 10 native populations. (**b**) Dry weight of stem, root, and leaf and total dry weight of 10 invasive and 10 native populations. (**c**) Morphology of US01 and CN25 populations in a glasshouse. (**d**) Plant height and stem diameter of US01 and CN25 populations. Data represent means ± SD (*n* = 10 populations for (**a**,**b**) and *n* = 9 biological replicates for (**d**)). Asterisks indicate significant differences between IN and NA populations (*p* < 0.01).

**Figure 2 biology-12-01347-f002:**
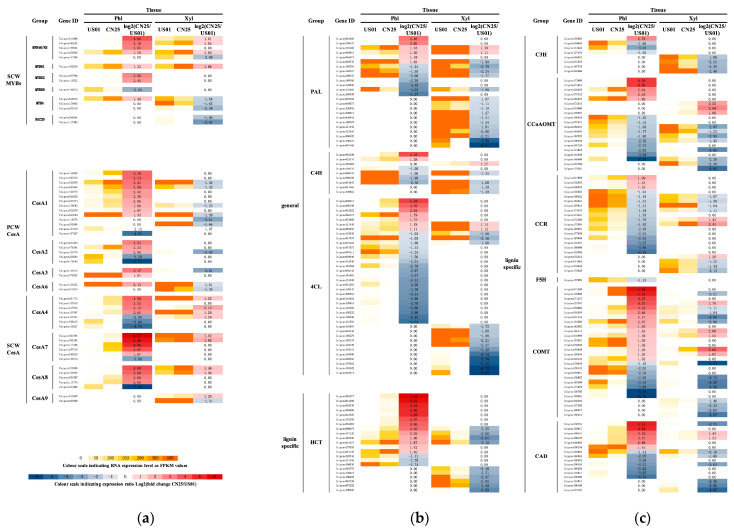
Expression of major genes involved in the process of SCW biosynthesis in US01 and CN25. (**a**) MYB transcription factor expression. (**b**) CesA and CSL expressions. (**c**) Phenylpropanoid pathway gene expression. The expression pattern of each uni-transcript is shown in six columns; columns 1, 2, 4, and 5 show the RNA expression level as normalized FPKMs across the phloems (columns 1, 2) and xylems (columns 4, 5) of the two populations, and columns 3 and 6 show the relative log2 (expression ratio) between CN25 and US01 across the phloem (column 3) and xylem (column 6). Phl, phloem; Xyl, xylem.

**Figure 3 biology-12-01347-f003:**
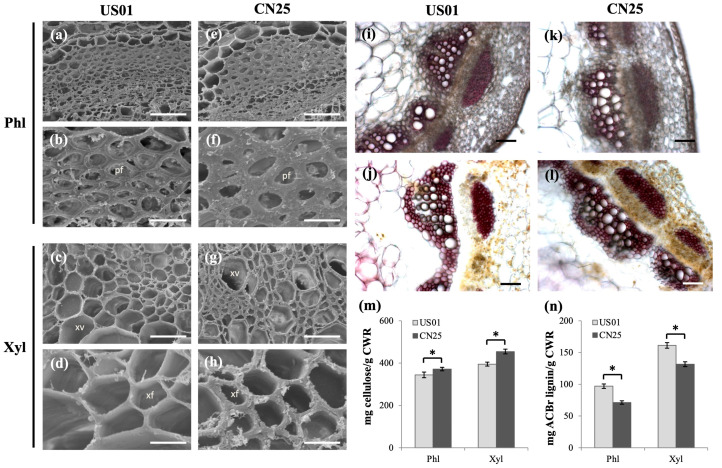
Comparison of stem anatomical and biochemical traits between invasive (IN) and native (NA) populations of *S. canadensis*. (**a**–**h**) Transmission electron micrographs of cross-sections of US01 (**a**–**d**) and CN25 (**e**–**h**) stems showing secondary wall thickening in the phloem (**a**,**b**,**e**,**f**) and xylem (**c**,**d**,**g**,**h**). (**i**,**k**) Wiesner staining of stem cross-sections of US01 **(i**) and CN25 (**k**) observed by light microscopy. Bars = 100 μm. (**j**,**l**) Maule staining of stem cross-sections of US01 (**j**) and CN25 (**l**) observed via light microscopy. Bars = 100 μm. (**m**) Cellulose content and (**n**) lignin content of stems from US01 and CN25. Asterisks indicate significant differences between the two populations (*p* < 0.01). pf, phloem fiber; xv, xylem vessel; xf, xylary fiber.

**Figure 4 biology-12-01347-f004:**
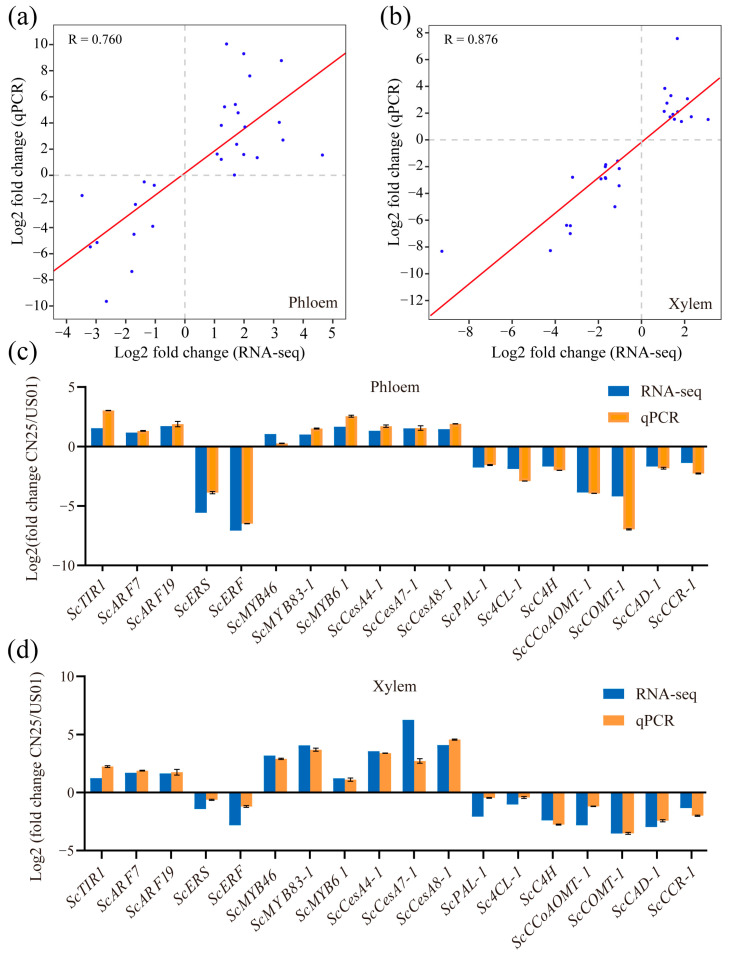
qRT-PCR validation of transcript levels evaluated using RNA-Seq in the phloem and xylem of *S. canadensis*. (**a**,**b**) Fold change correlation analyzed using the RNA-Seq platform (X-axis) with data obtained using real-time PCR (Y-axis) in the phloem and xylem of *S. canadensis.* (**c**,**d**) Comparison of the changes in expression of 18 selected genes detected using RNA-seq and qRT-PCR among CN25 and US01 in the phloem and xylem of *S. canadensis*. Error bars indicate standard deviations. Three biological replicates were used from each sample.

**Figure 5 biology-12-01347-f005:**
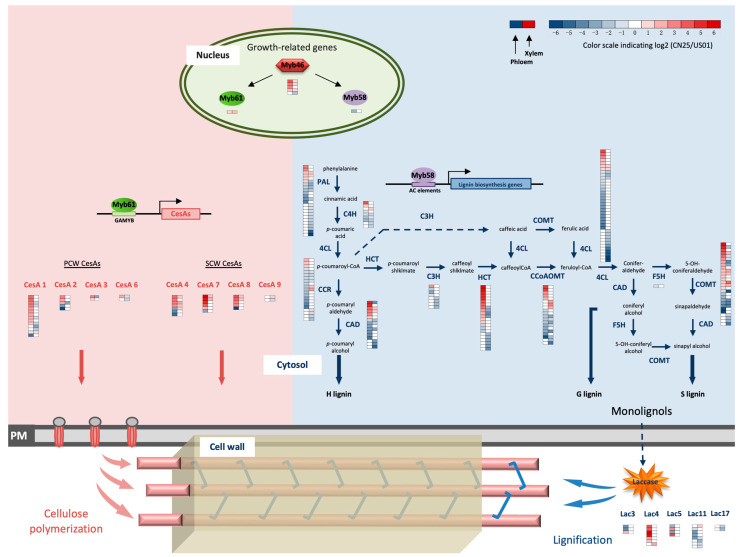
Comparison of the expression ratio of genes involved in the process of SCW biosynthesis between CN25 and US01 populations of *S. canadensis*. The expression ratio of each uni-transcript is shown in two grids, which represent the relative log2 (expression ratio) in the phloem (**left**) and xylem (**right**), respectively. Additionally, the heatmap (blue to red) shows the expression ratio between CN25 and US01 of each unigene.

**Figure 6 biology-12-01347-f006:**
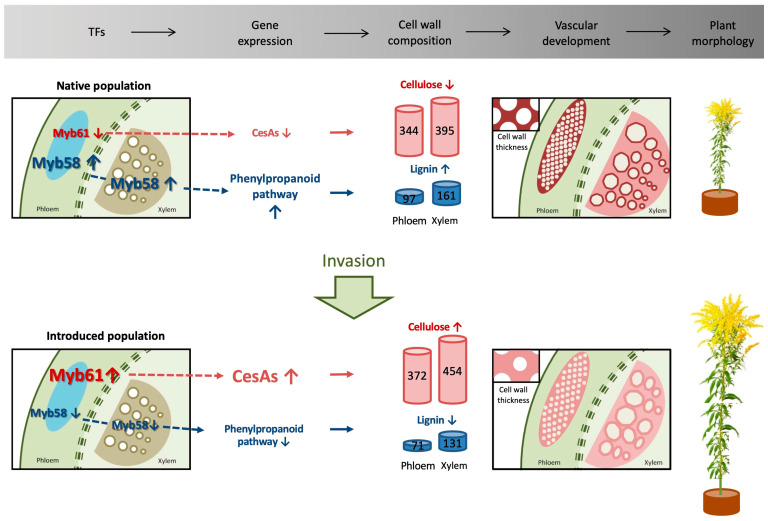
A model to describe the major transcriptional and physiological differences related to vascular tissue development between CN25 and US01 populations of *S. canadensis*. In the TFs and gene expression columns, the upward and downward arrows highlight the up- and down-regulated DEGs involved in the expression pathways, respectively. In the “cell wall composition” column, the upward and downward arrows indicate an increase or decrease in the composition of the cell wall, respectively.

**Table 1 biology-12-01347-t001:** Plant material.

Population NO.	Longitude	Latitude	Location
Invasive populations (IN)			
CN05	118.83	32.05	Jiangsu, CN
CN11	121.24	31.93	Jiangsu, CN
CN30	121.54	29.87	Zhejiang, CN
CN17	119.45	34.85	Jiangsu, CN
CN10	121.07	32.07	Jiangsu, CN
CN14	120.83	31.32	Jiangsu, CN
CN65	118.88	32.11	Jiangsu, CN
CN25	119.44	32.46	Jiangsu, CN
CN38	120.17	30.90	Anhui, CN
CN47	116.55	31.66	Anhui, CN
Native populations (NA)			
US01	79.23	37.15	765#, Altavista, VA, USA
US52	73.24	43.64	529# Fair Haven, VT, USA
US28	73.79	43.03	250# Maltca, NY, USA
US31	79.32	42.47	Arrowhead Dr. Dunkirk, NY, USA
US34	87.94	42.92	20th St. Oak Creek, WI, USA
US44	87.33	41.42	Crown Point Church, MO, USA
US59	94.27	45.54	2398 76th Ave St. Cloud, MN, USA
US60	95.88	45.61	US59&HWY28 Morris, MN, USA
US06	81.65	38.35	2 Cantley Dr. Charleston, WV, USA
US02	79.70	34.25	2591N Williston Rd. Florence, SC, USA

## Data Availability

The transcriptome data sources are presented in Appendix A. In cases where the occurrence data are unpublished, disclosure is withheld for privacy reasons. Requests for such data can be directed to the first author, Y.Z.

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
