# Peer review of "Tissue-Specific Transcriptomes in the Secondary Cell Wall Provide an Understanding of Stem Growth Enhancement in Solidago canadensis during Invasion"

_biology, 2023, doi:10.3390/biology12101347_

Round 1

Reviewer 1 Report

Zhang et al. present a novel analysis of tissue-specific transcriptomes in secondary cell wall between native and invasive Solidago canadensis. First, they collected samples from North America and East Asia (China) for native and invasive species. They first characterized the morphological difference between the two, and indicated the variation in biomass was mainly caused by the aboveground parts. Then, they performed RNA sequencing for the two groups of samples, conducted de novo transcriptome assembly, and subsequently performed differential gene expression analysis for phloem and xylem tissues. They found that genes including MYB transcription factors, cellulose biosynthesis genes, and lignin-related genes are differentially expressed between the two populations in phloem and xylem, which is subsequently validated by qRT-PCR. They also conducted anatomical and biochemical analyses for the two populations. Finally, the authors proposed a scheme for the differential gene expression and regulation between native and invasive species.    This manuscript generally rises to a fair standard of evidentiary proof and a robust standard of methodological detail.  The work presented is high quality and will be relevant to those that work in the fields of invasive species Solidago canadensis. 

Some specific comments: 1. In line 187, the authors are referring IN for invasive population, while in line 215, it is introduced. Please clarify.  2. In line 303, the authors is citing Figure 5B which does not exist. Please clarify  3. Is there any hypothesis for the unexpectedly lower proportion of dignified issued in CN25 compared to US01 (line 359)? Could you provide additional insights? 

There are a few typos in the paper that need to be corrected. Meanwhile, I recommend that the authors rephrase some sentences to make them look more fluent and clear. 

Author Response

Some specific comments:

  1. In line 187, the authors are referring IN for invasive population, while in line 215, it is introduced. Please clarify. 

Response: Thanks for the reviewer’s kind suggestion. According to his/her advices, it has been corrected in “invasive population” which can be found in Line 269, Page 6.

  1. In line 303, the authors is citing Figure 5B which does not exist. Please clarify 

Response: We are sorry for the mistake. The correction has been made. And the detailed revision can be found in Line 391 Page 9.

  1. Is there any hypothesis for the unexpectedly lower proportion of lignified issued in CN25 compared to US01 (line 359)? Could you provide additional insights? 

Response: Thanks for the valuable comments. In this study, we found that the lignin biosynthesis genes were downregulated in invasive population of S.canadensis, accompanied by a significant decrease in stem lignin content. We determine that this change will have a certain impact on the growth of stem, according to previous research, it was shown that lignification of secondary wall would formation of cross-linking among cell wall polymer which imparts ‘waterproofing’ capacity as well as mechanical strength, rigidity and environmental protection. However, the polymers also reduce cell wall extensibility which would hinder the elongation of cells. Therefore, we assume that the reduced lignin content of the invasive population contributes to the increase in stem size. We have tried to provide this insight in the discussion. The detailed revision can be found in Line 591-597, Page 14.

Comments on the Quality of English Language

Response: English editing has been made.

There are a few typos in the paper that need to be corrected. Meanwhile, I recommend that the authors rephrase some sentences to make them look more fluent and clear. 

Response: Thanks for the reviewer’s kind suggestion. Correction has been made.

Reviewer 2 Report

The manuscript by Zhang et al. presents a transcriptomic analysis of genes involved in the process of secondary cell wall biosynthesis in the invasive plant Solidago canadensis. The authors compared gene expression in the phloem and xylem between native and invasive populations of S. canadensis to study the mechanism of its stem growth enhancement during invasion.

Unfortunately, the manuscript has many “technical” shortcomings that do not allow to objectively evaluate its scientific significance. There are serious English issues, and I did not understand the meaning of some phrases. In addition, there are many other inaccuracies in the manuscript; I had an impression that the authors did not proofread it before submission. I suggest the authors to edit the text carefully.

Comments:

1)    Title: “Tissue specific transcriptomes in secondary cell wall understating the enhancement of growth ability…”. Did the authors mean “understanding”? In this case, the verb is missing. “Provide understanding”?

2)    Line 17: “regulated by MYB TFs…” According to the journal template, the Simple Summary should not include any abbreviations or any technical terms without explanations.

3)    Line 66: «Secondary» should be lowercase.

4)    Line 102: «Together these analyses revealed the crosstalk» Should be «… revealed that the crosstalk…»

5)    Line 109: «sampling multiple ramets of the same individual genet.» The meaning of this sentence is unclear.

6)    Line 130: «night individual plants per population». Nine individual plants?

7)    Line 134: «Each experiment was repeated three times and performed with 10 plants for each replicate.». If the authors mention nine plants in line 130, how can there be ten?

Moreover, in the legend to Figure 1 it is written about 10 biological replicates.

8)    Lines 140-144: «then rinse 3-4 times with distilled water until the water solution stays clear. Discard the water. Quickly add 15% HCl until the deep brown color is discharged from the sections, pipette out all the 15% HCl solution and immediately add concentrated ammonium hydroxide solution.» It looks like copied from the manufacturer`s protocol; the authors should have at least put it in the past tense.

9)    Line 150: «were immediately stored in in liquid N2». What is liquid N2?

10) Line 152: «Total RNA of each sample with three biological replicates was isolated using TRIzol® Reagent (Invitrogen) according to the manufacturer’s protocol.» There are many versions of TRIzol based protocols, please indicate the specific one (or provide the reference).

11) The names of the genes should be italicized throughout the manuscript. Text in the “Results” subsections should be divided into paragraphs to make it easier to read.

12) Line 189: «However, PH, SD and LA...» These abbreviations should be spelled out in the text above.

13) Line 189: «IN populations were higher than that of NA populations with 105.9% (p < 0.01), 23.3% (p < 0.01) and 159.4% (p < 0.01),» In my opinion, the authors should say, for example, that the IN plants were 1,5 or 2.5 times higher than the NA plants. 159.4% looks weird.

14) Line 218: «Plants grown of US01 and CN25 populations in the growth chamber». In the “Materials and Methods” section, the authors mention only greenhouse (glasshouse), not the growth chamber.

15) Line 220: «for D» For panel (d)?

16) Line 235: «amino acid transport and metabolish». Should be «metabolism».

17) Line 239: «molecular function» category. «molecular function» should be in quotes.

18) Line 245: «The expression ratio of each uni-transcript is shown on six grids: The heat map (white to orange, grids 1,2,4 and 5)». I suppose that the authors mentioned columns; however, I see only three columns for each tissue: columns 1 and 2 range from white to orange, and column 3 ranges from blue to red. Please, provide the correct legend to Figure 2.

19) Line 247: «each differentially expressed MYB». MYB genes are presented only in panel A.

20) Line 271: «Serval MYB transcription factors». Several?

21) Lines 291-294: «the majority of these genes were up-regulated in both phloem and xylem of CN25, such as galacturonosyltransferase (GAUT), fructokinase (FRK)». Should be: “galacturonosyltransferase gene” or “gene encoding galacturonosyltransferase”.

22) Figure 4 legend: Please italicize “S. canadensis” .

23) Line 303: “Figure 5B” – is it correct?

24) Line 315: «lignin Content». Should be lowercase.

25) Line 369: “30 genes were first randomly selected for qRT-PCR assays (Table S12)”. I did not understand how the 18 genes presented in Figure 4 were selected from these thirty.

26) Line 398: “28 independent genes were randomly selected”. Figure 4 shows 18 genes.

27) Line 389: “This finding is consistent with that of Szymura (2015)”. The reference to this article is missing from the text.

28) Line 427: “Huang (2015) demonstrated that MYB61 could bind to…”. The reference to Huang et al., 2015 is missing either.

29) Line 477: “Expression ration” (ratio?)

30) Figure 5: Is the term “cellulosization” correct?

31) Line 505: “arrows highlight the up-DEGs and down-DEGs”. Should be “up- and down- regulated DEGs”.

Author Response

Comments: 

  • Title: “Tissue specific transcriptomes in secondary cell wall understating the enhancement of growth ability…”. Did the authors mean “understanding”? In this case, the verb is missing. “Provide understanding”?

Response: Thanks for the reviewer’s kind suggestion. Correction has been made.

  • Line 17: “regulated by MYB TFs…” According to the journal template, the Simple Summary should not include any abbreviations or any technical terms without explanations.

Response: Thanks for the reviewer’s kind suggestion. Correction has been made.

  • Line 66: «Secondary» should be lowercase.

Response: Correction has been made.

  • Line 102: «Together these analyses revealed the crosstalk» Should be «… revealed that the crosstalk…»

Response: Correction has been made.

  • Line 109: «sampling multiple ramets of the same individual genet.» The meaning of this sentence is unclear.

Response: Thanks for the questions. Due to the fact that S.canadensis can expand its population through asexual reproduction, it is advisable to maintain a distance of at least 10m between different populations during sampling to avoid guessing different clones of the same genotype. The correction has been made. and the detailed revision can be found in Line 165, Page 3.

  • Line 130: «night individual plants per population». Nine individual plants?

Response: Correction has been made.

7)    Line 134: «Each experiment was repeated three times and performed with 10 plants for each replicate.». If the authors mention nine plants in line 130, how can there be ten?

Moreover, in the legend to Figure 1 it is written about 10 biological replicates.

Response: We are sorry for the mistake. In this study each experiment was performed with 9 plants. Correction has been de, and the detailed revision can be found in Line 199, Page 3, and in Line 318, Page 6.

  • Lines 140-144: «then rinse 3-4 times with distilled water until the water solution stays clear. Discard the water. Quickly add 15% HCl until the deep brown color is discharged from the sections, pipette out all the 15% HCl solution and immediately add concentrated ammonium hydroxide solution.» It looks like copied from the manufacturer`s protocol; the authors should have at least put it in the past tense.

Response: Thanks for the reviewer’s kind suggestion. Editing has been made.

  • Line 150: «were immediately stored in in liquid N2». What is liquid N2?

Response: Thanks for the questions. It’s “liquid nitrogen”, correction has been made.

  • Line 152: «Total RNA of each sample with three biological replicates was isolated using TRIzol® Reagent (Invitrogen) according to the manufacturer’s protocol.» There are many versions of TRIzol based protocols, please indicate the specific one (or provide the reference).

Response: Thanks for the reviewer’s kind suggestion. Correction has been made, and the detailed revision can be found in Line 216, Page 4.

  • The names of the genes should be italicized throughout the manuscript. Text in the “Results” subsections should be divided into paragraphs to make it easier to read.

Response: Thanks for the reviewer’s kind suggestion. Editing has been made in the whole text.

  • Line 189: «However, PH, SD and LA...» These abbreviations should be spelled out in the text above.

Response: Thanks for the reviewer’s kind suggestion. These abbreviations had been spelled out in Materials and Methods, the detailed revision can be found in Line 185-188, Page 4.

  • Line 189: «IN populations were higher than that of NA populations with 105.9% (p < 0.01), 23.3% (p < 0.01) and 159.4% (p < 0.01),» In my opinion, the authors should say, for example, that the IN plants were 1,5 or 2.5 times higher than the NA plants. 159.4% looks weird.

Response: Thanks for the reviewer’s kind suggestion. This section has been rewritten.

  • Line 218: «Plants grown of US01 and CN25 populations in the growth chamber». In the “Materials and Methods” section, the authors mention only greenhouse (glasshouse), not the growth chamber. 

Response: Correction has been made.

  • Line 220: «for D» For panel (d)?

Response: Correction has been made.

  • Line 235: «amino acid transport and metabolish». Should be «metabolism».

Response: Correction has been made.

  • Line 239: «molecular function» category. «molecular function» should be in quotes.

Response: Correction has been made.

  • Line 245: «The expression ratio of each uni-transcript is shown on six grids: The heat map (white to orange, grids 1,2,4 and 5)». I suppose that the authors mentioned columns; however, I see only three columns for each tissue: columns 1 and 2 range from white to orange, and column 3 ranges from blue to red. Please, provide the correct legend to Figure 2.

Response: Thanks for the reviewer’s kind suggestion. Correction has been made, and the detailed revision can be found in Line 360-364, Page 7.

  • Line 247: «each differentially expressed MYB». MYB genes are presented only in panel A.

Response: Correction has been made.

  • Line 271: «Serval MYB transcription factors». Several?

Response: Correction has been made.

  • Lines 291-294: «the majority of these genes were up-regulated in both phloem and xylem of CN25, such as galacturonosyltransferase (GAUT), fructokinase (FRK)». Should be: “galacturonosyltransferase gene” or “gene encoding galacturonosyltransferase”.

Response: Correction has been made.

  • Figure 4 legend: Please italicize “S. canadensis” .

Response: Correction has been made.

  • Line 303: “Figure 5B” – is it correct?

Response: We are sorry for the mistake. The correction has been made. And the detailed revision can be found in Line 391 Page 9.

  • Line 315: «lignin Content». Should be lowercase.

Response: Correction has been made.

  • Line 369: “30 genes were first randomly selected for qRT-PCR assays (Table S12)”. I did not understand how the 18 genes presented in Figure 4 were selected from these thirty.

Response: Thanks for question. We have selected 4 pathways related to plant growth and biomass allocation which were enriched in different expressed analysis (Table S4 and S5). these 18 gene including 5 genes related to plant hormone signal transduction; 3 genes related to MYB transcription factor; 3 genes related to Starch and sucrose metabolism specially SCW biosynthesis; and 7 genes related to phenylpropanoid biosynthesis. We tried to illustrate this point in the article, and the detailed revision can be found in Line 508-515 Page 10.

  • Line 398: “28 independent genes were randomly selected”. Figure 4 shows 18 genes.

Response: Correction has been made.

  • Line 389: “This finding is consistent with that of Szymura (2015)”. The reference to this article is missing from the text.

Response: Thanks for question. This reference can be found in reference No.31, Line 804, page 16.

  • Line 427: “Huang (2015) demonstrated that MYB61 could bind to…”. The reference to Huang et al., 2015 is missing either.

Response: Thanks for question. This reference can be found in reference No.39, Line 821, page 16.

  • Line 477: “Expression ration” (ratio?)

Response: Correction has been made.

  • Figure 5: Is the term “cellulosization” correct?

Response: Correction has been made.

  • Line 505: “arrows highlight the up-DEGs and down-DEGs”. Should be “up- and down- regulated DEGs”.

Response: Thanks for the reviewer’s kind suggestion. The correct figure 5 has been replaced, and the detailed revision can be found in Line 655, Page 13.

Reviewer 3 Report

Large number of data, interesting approach. However, there are many points need to addressed before acceptance.

Firstly, paper need to be re-adjusted to fit senrtence sens and structure with correct grammar.

Few example (but there are much more):

Lines 9-10: please, fit both part of the sentences.

Lines 11: “few studies involved” ?

Lines 15- 18: complicated sentences, part did not fit perfectly.

Line 33: “process involved were differentially expressed” ?? Process can not expressed. I can predict what do you mean, but you need to explain it correctly.

Lines 34- 37: long sentience with non-fitting part, indeed.

Secondly, authors

Lines 53-54: highly questionable point: first, one need to have “source” of vascular system, and large number of photosynthetic cell. It mean cell division and origin formation is the primary.

Vascular development and increasing in vascular density without increasing the number of mesophyll can not have a positive effect.

 Line 119: can you provide volume of the soil per plant/pot?

Line 121: day length, light?

Line 129: during mean different time point as kinetics. Here you mention only one time point.

Line 145: “Carl Zeiss microimaging GmbH” ?? GmbH = company.

Line 150: “stored in in” ??

Line 185: “plant high” ???

Line 188: “RL”?? Root Length??

Fig 1 a – Y axis is not plant growth. It is plant organs size. Please, try to use variable scale for Y axis.

Line 195: “during invasion” ? Here you present data of the growth, not during invasion.

Line 202: “the mechanism of stem growth ability evolution of IN population

of S. canadensis at transcriptional level” - please, re-formulate. In your case you did not study growth ability since you study results of growth. These data have nothing common with griowth ability, but reflected rather large stem size, more secondary cell wall etc.

For the mechanism you need to study a kinetics 8day 5, 10, 20, 40, 60 as example. Your question must be at which stage staring divergence in growth between two populations? Which process is affected primary? Etc. These data can tell your about the reason, here your data reflected final results: more xylem, more phloem, more mesophyll. As one of the possible reason of such differences can be a local hormonal signaling. All the rest differences can be just a results of initial changes in this signaling, indeed.

Line 343: “To verify the effect of the above transcription level differences on stem growth” ? This is not correct. Here you study link of transcriptional level with results of large stem size. This is nothing to do with mechanism how large stem formed!

Line 403: “growth ability” ???

Line 503: “vascular tissues development” = vascular tissue abundance.

As conclusion, I would suggest to carefully check grammar and sentences structure, term meaning and mention in discussion that your numerous and interesting data reflected end point, not the primary mechanism.

Primary mechanism require different design of experiment.

require re/arrangements.

Author Response

Comments and Suggestions for Authors

Large number of data, interesting approach. However, there are many points need to addressed before acceptance.

Firstly, paper need to be re-adjusted to fit senrtence sens and structure with correct grammar.

Response: Thanks for the reviewer’s kind suggestion. According to his/her advices, sentence sense and structure in the paper had been readjusted.

Few example (but there are much more):

Lines 9-10: please, fit both part of the sentences.

Response: Correction has been made.

Lines 11: “few studies involved” ?

Response: Correction has been made.

Lines 15- 18: complicated sentences, part did not fit perfectly.

Response: Correction has been made.

Line 33: “process involved were differentially expressed” ?? Process can not expressed. I can predict what do you mean, but you need to explain it correctly.

Response: Correction has been made.

Lines 34- 37: long sentience with non-fitting part, indeed.

Response: Correction has been made.

Secondly, authors

Lines 53-54: highly questionable point: first, one need to have “source” of vascular system, and large number of photosynthetic cell. It mean cell division and origin formation is the primary.

Response: Thanks for the questions. This sentence has been rewritten to propose the impact of vascular tissue development on plant morphology and function. This point can be found in Line 100, Page 2.

Line 119: can you provide volume of the soil per plant/pot?

Response: Thanks for the reviewer’s kind suggestion. This information was given in this revised version and the detailed revision can be found in Line 269, Page 12.

Line 121: day length, light?

Response: There may be a misunderstanding here that I did not find the word “day length” in the text.

Line 129: during mean different time point as kinetics. Here you mention only one time point.

Response: Thanks for the reviewer’s kind suggestion. Correction has been made.

Line 145: “Carl Zeiss microimaging GmbH” ?? GmbH = company.

Response: Correction has been made.

Line 150: “stored in in” ??

Response: Correction has been made.

Line 185: “plant high” ???

Response: Correction has been made.

Line 188: “RL”?? Root Length??

Response: Correction has been made.

Fig 1 a – Y axis is not plant growth. It is plant organs size. Please, try to use variable scale for Y axis.

Response: Thanks for the reviewer’s kind suggestion. The correct figure 1 has been replaced, and the detailed revision can be found in Line 313, Page 6.

Line 195: “during invasion” ? Here you present data of the growth, not during invasion.

Response: Thanks for the questions. Here, we compare the differences between IN and NA populations in order to demonstrate the morphological and physiological changes of S.canadensis during the invasion process. Therefore, the term “during invasion” refers to the changes that occur after the invasion.

Line 202: “the mechanism of stem growth ability evolution of IN population

of S. canadensis at transcriptional level” - please, re-formulate. In your case you did not study growth ability since you study results of growth. These data have nothing common with griowth ability, but reflected rather large stem size, more secondary cell wall etc.

For the mechanism you need to study a kinetics 8day 5, 10, 20, 40, 60 as example. Your question must be at which stage staring divergence in growth between two populations? Which process is affected primary? Etc. These data can tell your about the reason, here your data reflected final results: more xylem, more phloem, more mesophyll. As one of the possible reason of such differences can be a local hormonal signaling. All the rest differences can be just a results of initial changes in this signaling, indeed.

Line 343: “To verify the effect of the above transcription level differences on stem growth” ? This is not correct. Here you study link of transcriptional level with results of large stem size. This is nothing to do with mechanism how large stem formed!

Response: The above two questions are similar, so I will answer them together. The comments of the reviewers are very valuable, and we also recognize that the results in this article are more focused on the "results" of growth rather than the "ability" of growth. Therefore, we have made relevant modifications to the viewpoint of the entire paper, more reflecting the changes in growth after successful invasion of S.canadensis, such as morphology, biomass, etc. In fact, this feature is also crucial for invasive organisms to enhance their competitiveness. Thanks for the reviewer’s valuable suggestion!

Line 403: “growth ability” ???

Response: Correction has been made.

Line 503: “vascular tissues development” = vascular tissue abundance.

Response: Thanks for the questions. Here, the differences between the two populations were not only in vascular tissue abundance but also in developmental such as the degree of vessel development and lignification. We also try to present this developmental difference in the schematic diagram. Therefore, we prefer to use “development”.

As conclusion, I would suggest to carefully check grammar and sentences structure, term meaning and mention in discussion that your numerous and interesting data reflected end point, not the primary mechanism.

Primary mechanism require different design of experiment.

Response: Thanks for the reviewer’s kind suggestion. Editing has been made in the whole text.

Round 2

Reviewer 2 Report

During the revision, the text of the manuscript was significantly improved, and now it reads well. It is obvious that the data analysis has been done correctly.

However, some corrections still need to be introduced. 

1)    Please include the details of qRT-PCR in the “Materials and methods” section: provide the name of the reference gene for data normalization; specify a fluorescent dye and the other reagents used; specify a type of the qPCR machine.

2)    In the revised version of the manuscript, Figure 2 has much lower resolution compared to the initial version. Please, improve the resolution of this figure.

3)    There are no "A" and "B" labels in Figure S5.

4)    Line 314: Should be “invasive (IN)”, not “introduced”.

5)    Line 16: The abbreviation “SCW” should spelled out when first mentioned.

6)    Line 34: “Based on a comprehensive analysis that relating of genes expression profiles on SCW deposition and vascular tissue development, we revealed that the variation of SCW biosynthesis genes expression during successful invasion of S.canadensis”. This sentence needs some improvement to ensure clarity.

7)    Line 144: “Then quickly add 15% HCl until the deep brown color of the sections disappeared, pipette out all the 15% HCl solution and immediately add a concentrated ammonium hydroxide solution.” Please rewrite this description in the past tense.

Author Response

Thank you very much for your valuable suggestions, which makes this paper more professional and clearer. Based on your suggestion, we have made correction one by one. The details are as follows:

  • Please include the details of qRT-PCR in the “Materials and methods” section: provide the name of the reference gene for data normalization; specify a fluorescent dye and the other reagents used; specify a type of the qPCR machine. 

Response: Thanks for the reviewer’s kind suggestion. This part had been added in Materials and Methods, the detailed revision can be found in Line 254, Page 4, of the revision mode.

  • In the revised version of the manuscript, Figure 2 has much lower resolution compared to the initial version. Please, improve the resolution of this figure.

Response: Thanks for the reviewer’s kind suggestion. The figure 2 in high resolution has been added. We have also uploaded the tiff format of all 6 figures in the attachment.

  • There are no "A" and "B" labels in Figure S5.

Response: Correction has been made.

  • Line 314: Should be “invasive (IN)”, not “introduced”.

Response: Correction has been made.

  • Line 16: The abbreviation “SCW” should spelled out when first mentioned.

Response: Correction has been made.

  • Line 34: “Based on a comprehensive analysis that relating of genes expression profiles on SCW deposition and vascular tissue development, we revealed that the variation of SCW biosynthesis genes expression during successful invasion of S.canadensis”. This sentence needs some improvement to ensure clarity.

Response: Thanks for the reviewer’s kind suggestion. According to his/her advices, this sentence has been improved.

  • Line 144: “Then quickly add 15% HCl until the deep brown color of the sections disappeared, pipette out all the 15% HCl solution and immediately add a concentrated ammonium hydroxide solution.” Please rewrite this description in the past tense.

Response: Correction has been made.

Reviewer 3 Report

Line 38: development is a process, you demonstrated presence/abundance.

Lines 295- 298: 5 times gene encoding in 1 sentence!!!!

Authors significantly improved text. However, in discussion it will be nececessary to add at least hypothesis about primary mechanism of invasion.

minor

Author Response

Thank you very much for your valuable suggestions, which makes the details of the paper more professional and clearer. Based on your suggestion, we have made correction one by one. The details are as follows:

Line 38: development is a process, you demonstrated presence/abundance.

Response: Thanks for the reviewer’s kind suggestion. According to the advices, this sentence has been rewritten.

Lines 295- 298: 5 times gene encoding in 1 sentence!!!!

Response: Thanks for the reviewer’s kind suggestion. Correction has been made.

Authors significantly improved text. However, in discussion it will be nececessary to add at least hypothesis about primary mechanism of invasion.

Response: Thanks for the reviewer’s kind suggestion. This part had been added in conclution, the detailed revision can be found in Line 713-719, Page 15, of the revision mode.
